# Polymorphisms in *rs2069845* are associated with IL-6 and soluble IL-6 receptor levels during total joint replacement

Kyle D. Anderson[ID][1], Bryan Dulion[1], John Wong[1], Niyati Patel[1], Anne DeBenedetti[2], Craig J. Della Valle[2], Ryan D. Ross[ID][1,2,3]*

1 Department of Anatomy and Cell Biology, Rush University Medical Center, Chicago, Illinois, United States of America, 2 Department of Orthopedic Surgery, Rush University Medical Center, Chicago, Illinois, United States of America, 3 Department of Microbial Pathogens and Immunity, Rush University Medical Center, Chicago, Illinois, United States of America

* Ryan_ross@rush.edu

## Abstract

As the number of patients undergoing total joint replacement (TJR) surgery increases, so does the number of revision surgeries. One driver of implant failure and subsequent revision surgery is peri-implant osteolysis, which is driven by inflammation-mediated bone loss. IL-6 is an inflammatory cytokine that is elevated during the peri-operative period. Early elevations in IL-6 levels have been linked to osteolysis development. The current study asked whether there is genetic contribution to the IL-6-related peri-operative inflammatory reaction to TJR surgery. Patients undergoing primary TJR (total hip or total knee) provided pre-operative and post-operative blood samples for measurement of the circulating levels of IL-6 and the soluble IL-6 receptor (sIL-6r), as well as evaluation of allele status of three single nucleotide polymorphisms (SNPs) linked to IL-6 or sIL-6r levels - rs2069845, rs2228145, and rs4537545. Circulating sIL-6r levels were associated with allele status in the rs2228145 SNP. More interestingly, allele status in the rs2069845 SNP was associated with the change in circulating IL-6 levels following TJR surgery. Specifically, patients with the A,A allele had increasing levels of IL-6, while those harboring the G,A allele had decreasing levels of IL-6. While implant survival was not assessed, the critical role of IL-6 in peri-implant osteolysis suggests that the rs2069845 allele may influence orthopedic implant success. rs2069845 polymorphisms may be a useful patient-specific marker of inflammatory response to TJR surgery.

## Introduction

The number of revision total hip and knee replacement surgeries performed in the U.S. is projected to grow to between 200,000 and 300,000 by the year 2030 [1]. The most common indications for revision surgery are implant instability and aseptic

**Data availability statement:** All relevant data are within the manuscript and its Supporting Information files

**Funding:** The research was supported by the Rush Scientific Leadership Council through the Searle Innovators Award to RDR and through the National Institutes of Health (NIH), National Institute of Arthritis Musculoskeletal and Skin (NIAMS) through T32AR073157 to KDA.

**Competing interests:** The authors have declared that no competing interests exist.

loosening [2], which are often attributed to the biological loss of implant fixation caused by particle-induced peri-implant osteolysis [3,4]. Despite advances in materials, such as ultra-high molecular weight polyethylene (UHMWPE) liners [5], aseptic loosening remains the primary cause of total hip revisions and the second leading cause of total knee revisions, as highlighted by recent registry analyses [6]. Peri-implant osteolysis progresses for years prior to diagnosis resulting in substantial bone loss, which makes revision surgeries more challenging [7,8]. Identifying patients at risk for aseptic loosening may allow for early intervention to halt or reverse bone loss, as has been demonstrated in preclinical model systems and clinical case-reports [9,10]. However, to date, no validated biomarkers have been established to diagnose osteolysis or subsequent aseptic loosening [11].

The pathophysiology of peri-implant osteolysis involves inflammation-induced bone loss caused by wear particles generated from prosthetic materials, such as polyethylene [12,13]. Therefore, it is not surprising that several cytokines have been proposed as late-stage biomarkers of implant failure caused by osteolysis, including interleukin-6 (IL-6), tumor necrosis factor (TNF), IL-1β, and IL-8 [11]. Recently, our lab has reported that elevated levels of IL-6 were among a panel of biomarkers able to prospectively discriminate total joint replacement patients that would eventually develop radiographically confirmed peri-implant osteolysis [14]. Perhaps more interestingly, pre-operative IL-6 levels (circulating levels measured prior to implant placement surgery) were equally able to distinguish the patients that would eventually develop radiographic osteolysis, suggesting that patient intrinsic factors exist that predispose patients to osteolysis development and these factors are likely related to elevated IL-6 signaling.

IL-6 is an inflammatory cytokine that is associated with osteoclastogenesis, leading to bone resorption and eventual loss of bone mass [15,16]. The mechanisms driving IL-6-induced osteoclastogenesis are complex (reviewed here [17]). IL-6 receptor signaling can occur by either *cis*- or *trans*-signaling mechanisms. *Trans*-signaling requires that IL-6 binds to the soluble IL-6 receptor (sIL-6r), this complex then requires the transmembrane glycoprotein 130 (gp130) to initiate intracellular signaling events. In contrast to *trans*-signaling, during *cis*-signaling IL-6 binds to membrane bound IL-6r rather than the soluble form. Several studies have noted that the induction of osteoclastogenesis requires the presence of both IL-6 and the soluble form of IL-6r [15,18,19], suggesting that *trans*-signaling is the predominate mode of action by which IL-6 signaling influences osteoclasts and bone resorption.

IL-6 induced osteoclastogenesis has been implicated in the pathogenesis of peri-implant osteolysis. Indeed, IL-6 is elevated in the peri-implant environment of failed orthopedic implants [20,21]. Further, single nucleotide polymorphisms (SNPs) in the *IL6* promotor region have been associated with implant lifespan [22,23] and severity of osteolysis [24], in support of our proposed patient intrinsic factors. The current study aimed to evaluate whether SNPs involved in the regulation of the IL-6 signaling pathway are associated with the early inflammatory response to total hip and knee replacement surgery. We focused on three SNPs previously linked to IL-6/

sIL-6r regulation. The selection of SNPs focused on our previous publication that noted that (1) higher post-operative levels of IL-6 were associated with osteolysis and (2) that patients that developed osteolysis had increasing IL-6 levels following surgery when compared to those that did not. Therefore, two SNPs were selected based on their link to circulating IL-6 or sIL-6r levels. The first SNP, rs4537545, occurs in the *IL6R* gene and has been implicated in the regulation of circulating levels of both IL-6 and sIL-6r [25–27]. rs2228145 (also referred to as rs8192284) is within the *IL6R* gene and has been linked to sIL-6r levels [28] and coronary heart disease risk [29]. rs2069845, which occurs in the *IL6* gene, was also included as it has been linked to IL-6 dependent inflammatory reactions. Specifically, leprosy patients with rarer rs2069845 genotypes are reported to an increased risk for and more severe inflammatory reactions [30]. We hypothesized that allele variance in these SNPs would be associated with changes in circulating IL-6 and sIL-6r in the peri-operative period of patients undergoing primary total joint replacement.

## Materials and methods

The study design was approved by the Rush University Institutional Review Board number protocol number 17061902-IRB01. A signed consent form was obtained from each patient providing permission for storage of biofluids for future research studies.

### A priori power analysis

Our study design was focused on evaluating whether SNPs impact how circulating IL-6 levels change following joint replacement surgery. Therefore, to power the study, we relied on the frequency of allele variations in the three proposed SNPs, which range in prevalence between 32–45% of the population (NCBI 1000 Genomes Browser) and our published IL-6 change data [14]. Power analysis was performed using the pre-surgical and post-surgical IL-6 levels for patients that would eventually develop osteolysis vs. those that did not develop osteolysis. From that data, the estimated sample sizes needed to detect a difference were calculated as 15 and 25, respectively, assuming a 3:1 sample to control distribution (based on the estimated 32% minor allele frequency). As we were most interested in the response in IL-6 to surgery, we also used the within-person IL-6 change data (i.e., post-surgical IL-6 levels minus pre-surgical levels), which led to a sample size estimate of 17 participants. We further validated our sample size estimate using the Quanto power calculation tool (version 1.2.4). Assuming a continuous outcome design, a gene only hypothesis, and the allele frequencies described above, we estimated that a sample size of 22 or more would allow us to detect an effect size ($R^2_G$ of 0.3 or greater). In total, 23 patients were consented and provided both pre and post-operative blood samples.

### Participant enrollment

A total of 34 patients receiving either primary total hip replacement (THR) or primary total knee replacement (TKR) surgeries with no prior history of inflammatory disease were consented at a pre-operative consultation to enroll in the present study. Enrollment occurred between March 2019 and August 2022. Participants were provided informed consent forms, which were signed in the presence of the study investigators. A total of 7 participants subsequently canceled their scheduled surgeries and an additional 4 participants were excluded due to missing pre- or post-operative samples. In total, 23 patients receiving primary total hip or knee replacements were included, thereby achieving our sample size.

Blood samples were collected by trained phlebotomists as part of the venipuncture service of the Rush Medical Laboratories. The average time between the pre-operative sample collection date and surgery was 10 (± 10) days and the average time between the post-operative collection data and surgery was 33 (± 16) days. Blood samples were collected in K2 EDTA tubes (BD) and centrifuged at 2,500 RPM for 15 minutes to separate plasma and buffy coats. The resulting phases were aliquoted into fresh tubes and stored at -80ºC.

### Cytokine quantification

Plasma samples were thawed to room temperature prior to analysis. The circulating levels of IL-6 and sIL-6R were assessed using commercially available ELISAs (Human IL-6 ELISA, BioLegend) (Human IL-6 receptor [soluble] Human ELISA Kit, ThermoFisher). Due to limited plasma volume obtained for some participants, this study could not obtain results for six samples of IL-6 (four pre-operative and two post-operative samples) and one sIL-6R (post-operative sample).

### Single polymorphism (SNP) isotyping

SNP allele status was assessed using qPCR (Applied Biosciences). Validated Taqman primers for the SNPs of interest – rs2069845, rs2228145, and rs4537545 - were purchased from ThermoFisher. PCR reactions were set up on a 96 well plate and all patient samples were tested in triplicate. The minor allele frequency (MAF) for our target SNPs were identified using the 1000genomics database: rs2069845 (MAF; global: G=0.2526; European: G=0.4334), rs2228145 (MAF; global: C=0.2931; European: C=0.3598), and rs4537545 (MAF; global: T=0.4491; European: T=0.3738). The resulting SNP allele frequencies were tested using the Hardy-Weinberg Equilibrium (HWE) test package in R-studio using the likelihood-ratio test (HW.lrtest). All three SNPs matched the expectations of the Hardy-Weinberg principle ($p > 0.05$).

An in silico analysis was performed to predict the impact of the three SNPs evaluated on the IL-6 and sIL-6R using the Genomad database from the Broad Institute (gnomad.broadinstitute.org). rs2228145 is a missense mutation, therefore we leveraged the PolyPhen and SIFT databases to calculate scores of 0.02 for PolyPhen and 0.08 for SIFT, indicating mutations that are considered benign and tolerated (benign), respectively. rs2069845 and rs4537545 are SNPs within introns and therefore to perform in silico predictions, we relied on the CADD database. The resulting C scores for rs2069845 and rs4537545 were 4.25 and 0.324, respectively. The results point to moderate benign impact for both SNPs on IL-6 and sIL-6R, respectively. Therefore, in total the result suggest that the SNPs evaluated are unlikely to alter the structure and function of IL-6 and sIL-6R.

### Statistical analysis

Prism (Version 8; GraphPad) and SPSS (Version 19.0; SPSS Inc.) software packages were used for plotting and data analysis, respectively. Prior to testing, data was evaluated for normality using the Shapiro-Wilk test. Roughly half of the datasets were not normally distributed and therefore a more conversative non-parametric approach was used to compare means in all subsequent analyses. In all three SNPs evaluated, the rarest of the allele frequencies was significantly underpowered, therefore we focused on comparing the two more common allele frequencies using a non-parametric Mann-Whitney U test. To evaluate level changes pre- and post-operatively, we used mixed-effects ANOVA models. The effects of allele status, time (pre- vs. post-operative) and the interaction of these two terms are presented.

## Results

### Patient aemographics based on SNP Allele

The total number of patients receiving total knee replacement (TKR) surgery was 12, while 11 received total hip replacement (THR) surgery. There were 7 males and 16 females overall, with an average age of 67.6 (±9.1) years at the time of surgery. Participant demographic data according to participant SNP allele status are presented in Table 1.

### Effects of rs2069845 allele status on peri-operative IL-6 and sIL-6r levels

Neither pre-operative nor post-operative IL-6 levels were affected by rs2069845 allele status (Fig 1). However, there was a significant allele by time interaction when comparing the two more common alleles, A,A and G,A, indicating that patients harboring these two alleles react differently to TJR surgery. Specifically, patients with the A,A allele had increasing IL-6

**Table 1. Demographics and SNP alleles for patients undergoing primary total joint replacement surgery.**

| SNP Allele Status (sample size, %) | Surgery (# THR vs. TKR) | Sex (n, % female) | Age at Surgery (mean, range) |
|---|---|---|---|
| *rs2069845* (HWE: p = 0.897) | | | |
| A,A | 5 THR | 5 (45%) | 67.5 (58-84) |
| (11, 47.8%) | 6 TKR | | |
| G,A | 4 THR | 9 (90%) | 67.7 (49-86) |
| (10, 43.4%) | 6 TKR | | |
| G,G | 2 THR | 1 (50%) | 67.5 (63-72) |
| (2, 8.7%) | 0 TKR | | |
| *rs2228145* (HWE: p = 0.299) | | | |
| C,A | 6 THR | 4 (31%) | 67.2 (49-84) |
| (13, 56.5%) | 7 TKR | | |
| A,A | 4 THR | 6 (75%) | 66.8 (58-86) |
| (8, 34.7%) | 4 TKR | | |
| C,C | 0 THR | 1 (50%) | 73 (49-84) |
| (2, 8.7%) | 2 TKR | | |
| *rs4537545* (HWE: p = 0.128) | | | |
| C,T | 9 THR | 5 (33%) | 66.0 (49-84) |
| (15, 65.2%) | 6 TKR | | |
| C,C | 2 THR | 1 (20%) | 69.0 (58-86) |
| (5, 21.7%) | 3 TKR | | |
| T,T | 0 THR | 2 (67%) | 73.0 (70-76) |
| (3, 13.0%) | 3 TKR | | |

HWE – p-value derived from the Hardy-Weinberg Equilibrium (HWE) analysis using the likelihood-ratio test in R-studio.

THR – total hip replacement.

TKR – total knee replacement.

levels post-operatively, while those with the G,A allele had decreasing IL-6 levels. There were no significant allele differences in the pre-operative sIL-6r levels. Patients with the G,A allele had significantly elevated post-operative sIL-6r levels when compared to those with the A,A allele. There were no significant allele, time, or allele by time interactions in the sIL-6r levels.

### Effects of rs2228145 on peri-operative IL-6 and sIL-6r levels

The pre-operative and post-operative IL-6 levels were not affected by rs2228145 allele status (Fig 2). Nor were there time or interaction effects detected when comparing the pre- and post-operative change. Both pre- and post-operative sIL-6r levels were significantly higher in patients with the C,A allele compared those with the A,A allele, which was confirmed by the significant allele effect noted when comparing both time points together (allele effect: p = 0.004).

### Effects of rs4537545 on peri-operative IL-6 and sIL-6r levels

rs2228145 allele status had near significant effects on both IL-6 and sIL-6r levels (Fig 3). The pre- and post-operative levels of IL-6 were higher in patients with the C,T allele, but the results were not statistically significant, likely due to the relatively few patients with the C,C allele (Fig 3). When both timepoints were compared in the ANOVA, the results showed a trend toward an allele effect (p = 0.077). Similar effects were noted in the levels of sIL-6r, which were not significant at either timepoint, but there was a trend for an allele effect (p = 0.061).

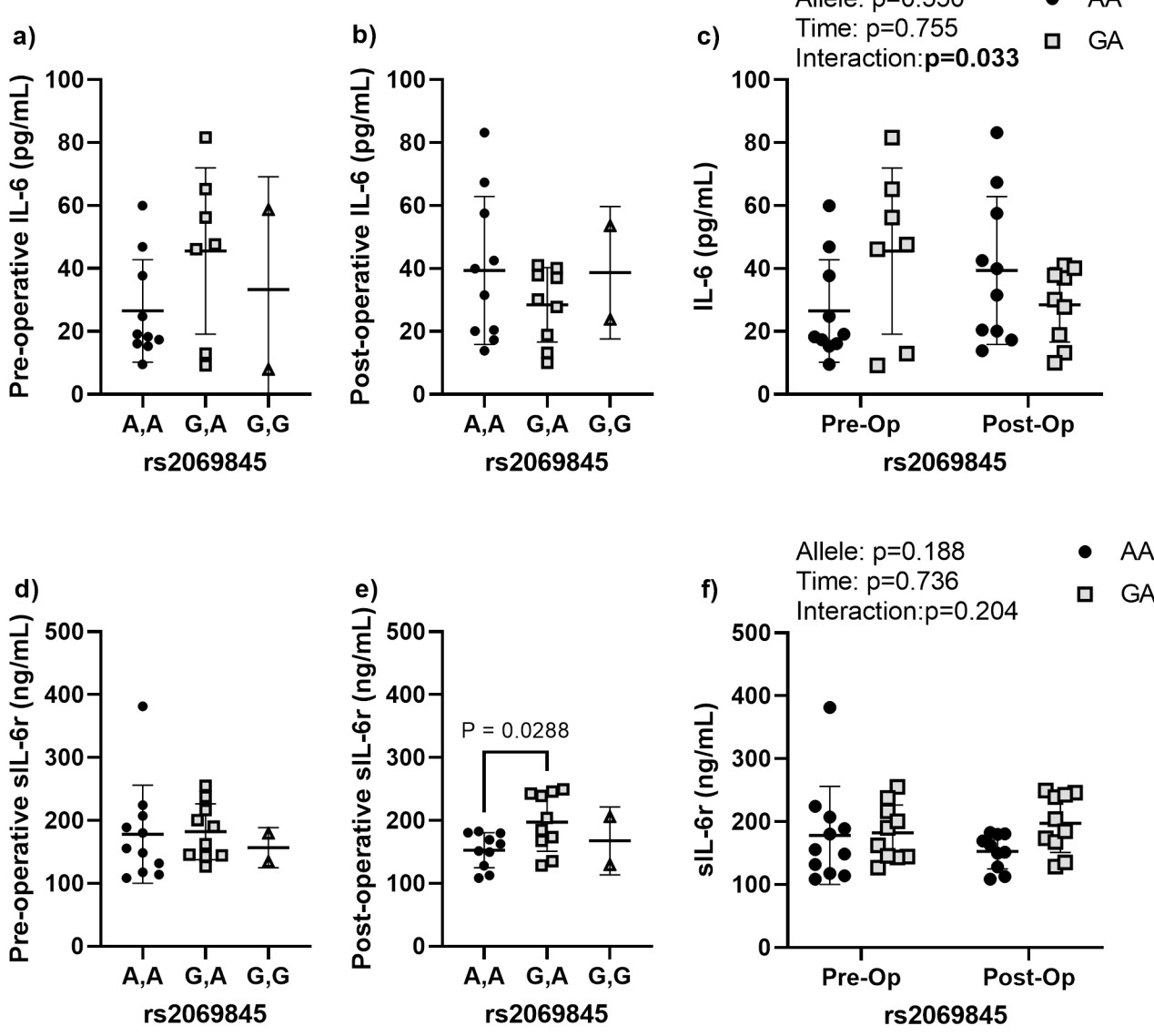

**Fig 1. Perioperative IL-6 and sIL-6r dynamics according to rs2069845 allele status. (a)** Pre-operative and **(b)** post-operative circulating IL-6 levels according to allele variant. **(c)** Comparison of the pre- and post-operative IL-6 levels according to allele variant for the two most prevalent alleles. **(d)** Pre-operative and **(e)** post-operative circulating sIL-6r levels according to allele variant. **(f)** Comparison of the pre- and post-operative sIL-6r levels according to allele variant for the two most prevalent alleles (A,A and G,A). Data are presented as individual measures with the mean and standard deviations. Significant pair-wise comparisons from non-parametric Mann-Whitney U tests are presented, when significant, as bars over the data in panels a, b, d, **e.** The results from non-parametric mixed effects comparisons of the effects of allele, time, and the allele by time interaction for the two most common alleles are presented as legends in panels c and **f.**

## Discussion

Peri-implant osteolysis and the subsequent loss of implant stability is one of the primary causes of total joint replacement failure [2–4,6]. There are currently no treatment options to halt or reverse peri-implant osteolysis progression. Osteolysis is generally only identified when a patient presents with pain or instability, at which point the only corrective action available is revision surgery. Revision surgeries for hip and knee replacements are more complex, costly, and have worse

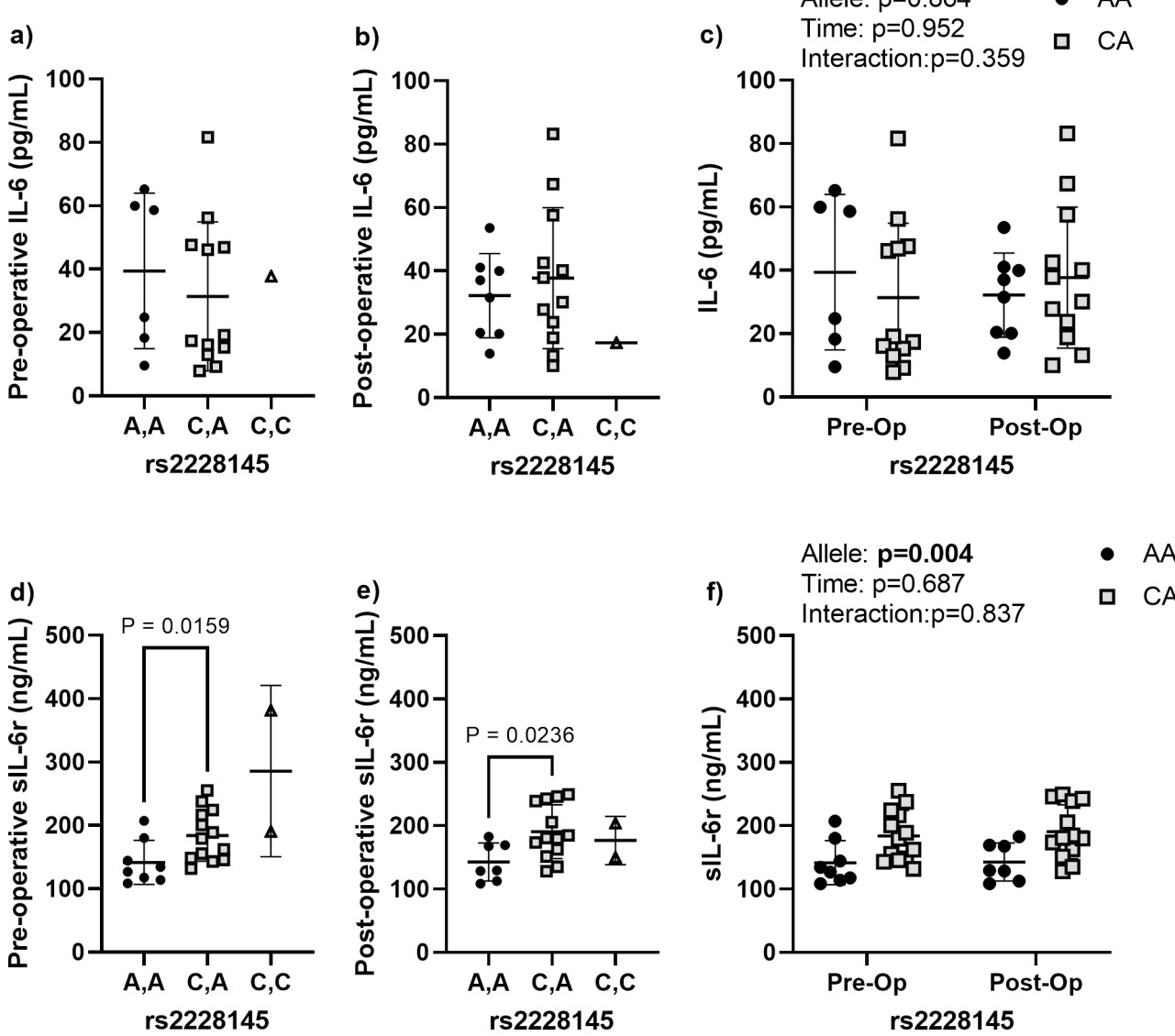

**Fig 2. Perioperative IL-6 and sIL-6r dynamics for rs2228145. (a)** Pre-operative and **(b)** post-operative circulating IL-6 levels according to allele variant. **(c)** Comparison of the pre- and post-operative IL-6 levels according to allele variant for the two most common alleles. **(d)** Pre-operative and (e) post-operative circulating sIL-6r levels according to allele variant. **(f)** Comparison of the pre- and post-operative sIL-6r levels according to allele variant for the two most common alleles (A,A and C,A). Data are presented as individual measures with the mean and standard deviations. Significant pair-wise comparisons from non-parametric Mann-Whitney U tests are presented, when significant, as bars over the data in panels a, b, d, e. The results from non-parametric mixed effects comparisons of the effects of allele, time, and the allele by time interaction for the two most common alleles are presented as legends in panels c and f.

outcomes when compared to primary surgeries [31–33], which are largely driven by the substantial bone loss associated with peri-implant osteolysis. Identifying risk factors for the development of osteolysis may allow for more routine clinical monitoring of at-risk patients, potentially allowing for early intervention prior to substantial bone loss and are critical for the future design of clinical trials for pharmaceutical treatments for osteolysis.

We have previously identified IL-6 levels as an early biomarker of subsequent osteolysis in one of the few longitudinal biomarker studies in total joint replacement patients [14]. Importantly the patients that ultimately developed osteolysis had

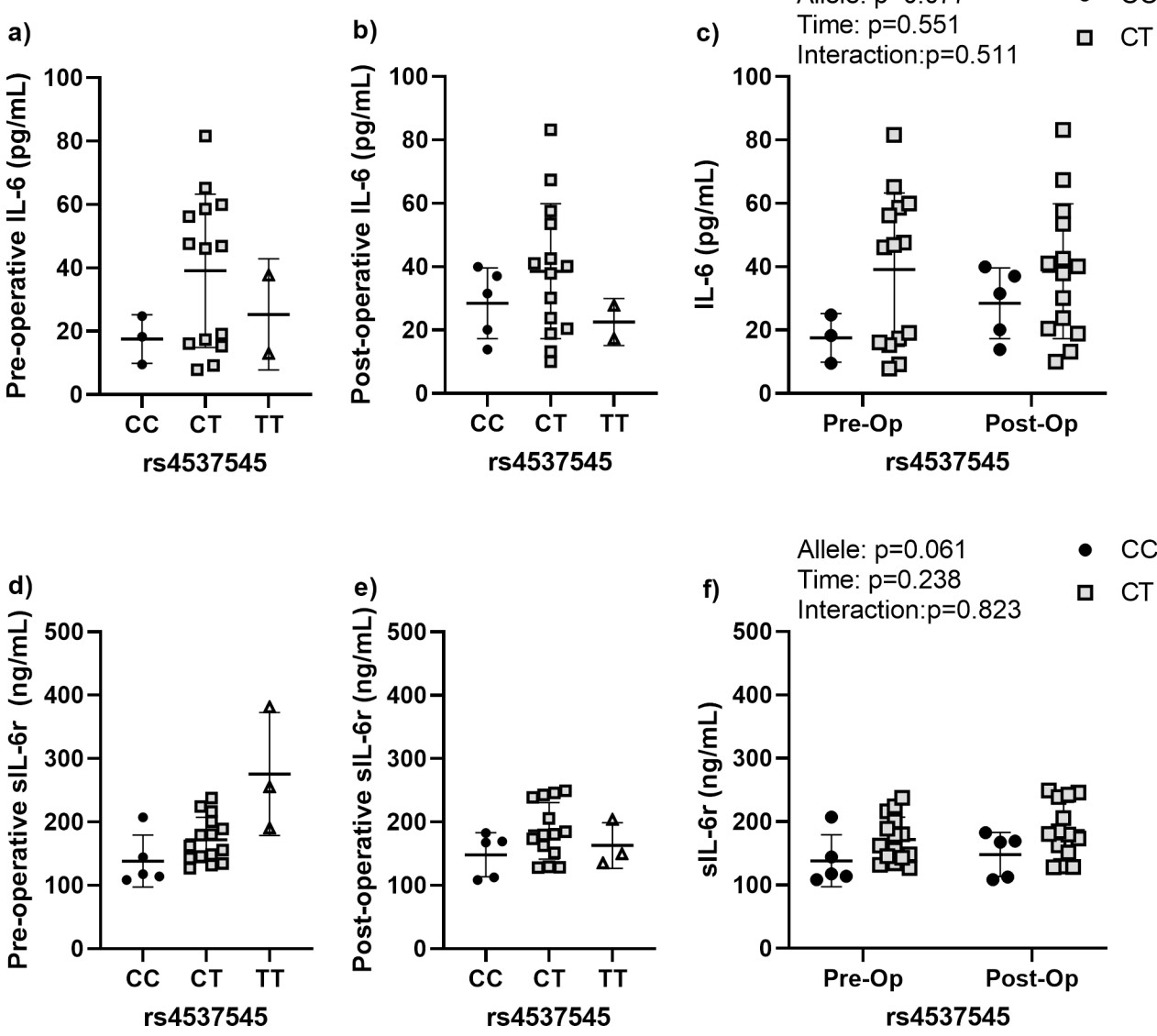

**Fig 3. Perioperative IL-6 and sIL-6r dynamics for rs4537545. (a)** Pre-operative and **(b)** post-operative circulating IL-6 levels according to allele variant. **(c)** Comparison of the pre- and post-operative IL-6 levels according to allele variant for the two most common alleles. **(d)** Pre-operative and **(e)** post-operative circulating sIL-6r levels according to allele variant. **(f)** Comparison of the pre- and post-operative sIL-6r levels according to allele variant for the two most common alleles (C,C and C,T). Data are presented as individual measures with the mean and standard deviations. Significant pair-wise comparisons from non-parametric Mann-Whitney U tests are presented, when significant, as bars over the data in panels a, b, d, e. The results from non-parametric mixed effects comparisons of the effects of allele, time, and the allele by time interaction for the two most common allele are presented as legends in panels c and f.

lower pre-operative IL-6 levels but higher post-operative levels than patients that did not develop osteolysis over the same period, suggesting a possible patient-intrinsic sensitivity to total joint replacement surgery and implant materials. The potential genetic contribution to osteolysis has been summarized by Jagga et al [34], which highlighted a variety of SNPs that have been linked to the biological response to implant wear particles or risk of osteolysis. Many of the SNP alleles that have been linked to joint replacement surgery outcomes are involved in the regulation of pro-inflammatory cytokines, such as IL-6.

IL-6, a major cytokine that controls acquired immunity and chronic inflammation [35], is elevated in the peri-implant tissues of implants that failed due to osteolysis [20,36–40]. IL-6 has also been studied as a circulating biomarker of osteolysis with mixed success [41], likely due to the fact that the majority of studies have investigated IL-6 levels at the end stage of osteolysis [42–45], after osteolytic lesions were detected. Longitudinal assessments of IL-6 levels following total knee replacement are reported to increase immediately post-operatively, reaching a peak as early as 48 hours after surgery [46]. These early changes in IL-6 are also reported in a variety of elective surgeries (summarized herein [47]), but prolonged elevation of IL-6 has been associated with adverse post-surgical complications, including mortality [48–50]. In the bone microenvironment, IL-6 induces the osteoblast production of RANKL, which subsequently activates osteoclast-mediated bone resorption [15,16]. Therefore, early and prolonged elevations in IL-6 levels likely contribute to sustained osteoclast activity, osteolysis development, and the loss of implant fixation.

Various single nucleotide polymorphisms (SNPs) influence IL-6 expression and circulating levels, potentially modulating inflammatory response. While we did not find any relationships between the evaluated SNPs and the pre- or post-operative levels of circulating IL-6, we note an interesting SNP by time interaction related to rs2069845 allele status. Specifically, patients harboring the A,A allele had the expected increase in IL-6 following TJR surgery, while those with the G,A, allele saw decreasing IL-6 levels post-operatively. Although preliminary, these results tend to suggest that the G,A may protect from inflammation-induced peri-implant bone loss. It is also possible that these findings represent a faster resolution of post-surgical inflammation, as leprosy patients harboring the G,A allele status in rs2069845 had significantly shorter times to reactional episodes [30]. This same study also noted no associations between rs2069845 allele status and the circulating levels of IL-6, which tends to suggest that this SNP is primarily involved in induced inflammatory reactions. Whether rs2069845 is associated with the risk of orthopedic implant failure is currently unknown, but efforts to reduce IL-6 in preclinical models of particle-induced osteolysis have noted significantly reduced osteoclastogenesis following IL-6 antibody treatment [51], suggesting that lower levels of IL-6 post-operatively may be protective.

sIL-6r is a marker of inflammation that potentiates the effects of IL-6 and polymorphisms in sIL-6r have likewise been associated with levels of both sIL-6r and IL-6 itself. Similar to previous publications, we found that rs2228145 was associated with the circulating levels sIL-6r [25,52,53]. Specifically, we noted that patients with the C,A allele in the rs2228145 had higher levels of sIL-6r both pre- and post-operatively. Patients with the C,T allele in rs4537545 also had elevated sIL-6r levels pre- and post-operatively but the results were not significant (p = 0.077). The C,T allele was also associated with elevated IL-6 levels both pre- and post-operatively, but again the elevation fell short of the significance threshold (p = 0.061). Together, these results suggest that variance in rs4537545 and rs2228145 alleles are likely to have higher baseline inflammation, but that their inflammatory environment is largely unaffected by TJR surgery.

IL-6 can have both direct and indirect effects on osteoclastogenesis. The direct effect of IL-6 on osteoclast precursor cells is somewhat controversial, with investigators reporting both enhanced [54] and inhibited [55] osteoclast differentiation in the presence of IL-6. The indirect effect involves IL-6/sIL-6r mediated signaling to neighboring osteoblast lineage cells. This response is dependent upon the expression of RANKL, a key cytokine, that is necessary for the differentiation of osteoclasts. RANKL has been evaluated previously as a marker for peri-implant osteolysis, however the results have been relatively mixed with the majority of studies finding no differences at the circulating level between stable and osteolytic joints [41]. High levels of RANKL have been reported in the peri-implant tissues surrounding osteolytic implant materials [56], which may indicate that locally produced, membrane-bound RANKL is the primary driver of osteoclast activation around joint replacement. Notably, a similar response is noted in ovariectomy (OVX)-induced bone loss, wherein the deletion of soluble or circulating RANKL did not prevent OVX-induced bone loss due to the high expression of RANKL in the local bone microenvironment [57].

A variety of patient-specific, or host, factors have been identified as contributing to hip or knee replacement failure. Some of these factors are potentially modifiable, such as high activity levels following surgery, while some are not, such as male sex at birth [58]. Osteoporosis [59] and diabetes mellitus [60] have also been reported to increase the risk for joint placement failure and suggest that proper management of comorbid conditions is critical to the success of orthopedic

implants. The current study investigated the genetic contribution to implant failure. Several previous studies have reported links between SNPs and osteolysis or other joint replacement failure mechanisms. Examples of the identified SNPs include SNPs in the *TNF* [61] and *IL1RA* [22] genes, which have been associated with increased risk for osteolysis development around total hip replacements, and SNPs in *IL1* have been linked to aseptic failure of both total hip and knee replacements [62]. Additionally, there have been SNPs in other regions of the *IL6* regulatory elements that have been linked to aseptic loosening of total hip implants [23,22]. Interestingly, immune cells from patients with previous osteolysis diagnosis express greater levels of these same cytokines (*IL1, IL6, TNFα*) in response to simulated wear particles [63], suggesting that these genetic changes may lead to an increased sensitivity to implant materials. Additional SNPs have been linked to peri-implant osteolysis around dental implants. A formal review of these studies is outside of the scope of this manuscript but are summarized here [34]. To our knowledge, none have evaluated SNPs in the IL-6 receptor *gp130* and peri-implant osteolysis. It is worth noting that SNPs in *gp130* have been linked to a variety of inflammatory diseases [64].

Our study is not without limitations. The sample size was limited, and although an a-priori power analysis was conducted to determine the required sample size, the study was underpowered to analyze the effects of rarer alleles. However, the collected data can inform and enhance the design of larger, more robust studies. Our study did not include a control population. While IL-6 levels following surgery have been reported previously, it isn't clear whether there is a normal range that is linked to better long-term performance. Therefore, it isn't clear what population would best serve as a control, so instead we relied on each individual's pre-surgical values to serve as an intra-individual control. Further, we do not know whether any of the patients recruited developed peri-implant osteolysis. Patients were recruited between 2018 and 2020 and because few patients ultimately develop osteolysis and those that do only present with osteolysis years after primary surgery, follow-up studies would take years to complete and require much larger sample sizes. However, as we have previously reported that early IL-6 levels are predictive of subsequent osteolysis [14], these data provide potential mechanistic insight linking a patient's genetics to the early IL-6 response to TJR surgery. Finally, the timing of post-operative sample collection ranged from 15 to 52 days after surgery. The post-operative sample collection occurred during the regularly scheduled post-operative in-person visit and was subject to patient availability. At least two studies have evaluated circulating IL-6 levels longitudinally following TJR surgery and reported that IL-6 levels are back to pre-operative levels two weeks after uncomplicated surgery [46,65]. Therefore, it is likely that despite the sample collection variability, all patients had recovered to a new baseline following the initial surgery-induced IL-6 spike, which is reported to occur within the first 48 hours.

## Conclusions

Peri-implant osteolysis is a rare complication following total joint replacement surgery that is driven by the inflammatory response to wear particles. Peri-implant osteolysis progresses silently and is generally only diagnosed after substantial peri-implant bone loss. Identifying risk factors can help guide clinical decision making prior to substantial bone loss. In the present study, we determined that SNPs related to the regulation of IL-6 and sIL-6r, particularly in the *rs2069845* SNP, are related to the inflammatory response to surgery. Due to the critical role of IL-6 in the regulation of osteoclast-induced bone loss, these results suggest that patient genetics contribute to the inflammatory response to total joint replacement surgery. Further work is needed to determine whether these early inflammatory reactions contribute to the development of peri-implant osteolysis and subsequent implant failure in the long-term.

## Supporting information

**S1 Data. A deidentified raw data set is attached to this manuscript, providing soluble IL-6 and sIL-6r levels, as well as SNP allele statuses.**
(XLSX)

## Acknowledgments

The authors thank the patients at Midwest Orthopaedics at Rush that volunteered to take part in this study. We also thank Dr. Vasili Karas and his research staff for their assistance in the recruitment of joint replacement patients.

## Author contributions

**Conceptualization:** Craig J. Della Valle, Ryan D. Ross.

**Data curation:** Kyle D. Anderson, Bryan Dulion, John Wong, Niyati Patel, Anne DeBenedetti, Ryan D. Ross.

**Formal analysis:** Kyle D. Anderson, Bryan Dulion, John Wong, Niyati Patel, Ryan D. Ross.

**Funding acquisition:** Ryan D. Ross.

**Investigation:** Kyle D. Anderson, Bryan Dulion, John Wong, Niyati Patel, Ryan D. Ross.

**Methodology:** Kyle D. Anderson, Anne DeBenedetti, Craig J. Della Valle, Ryan D. Ross.

**Project administration:** Anne DeBenedetti, Ryan D. Ross.

**Supervision:** Craig J. Della Valle, Ryan D. Ross.

**Writing – original draft:** Kyle D. Anderson, Ryan D. Ross.

**Writing – review & editing:** Kyle D. Anderson, Bryan Dulion, John Wong, Niyati Patel, Anne DeBenedetti, Craig J. Della Valle, Ryan D. Ross.

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
