## [Decision Letter · Decision Letter 0]

13 Dec 2024

PONE-D-24-46321Polymorphisms in rs2069845 are associated with IL-6 and soluble IL-6 receptor levels during total joint replacementPLOS ONE

Dear Dr. Ross,

Thank you for submitting your manuscript to PLOS ONE. After careful consideration, we feel that it has merit but does not fully meet PLOS ONE’s publication criteria as it currently stands. Therefore, we invite you to submit a revised version of the manuscript that addresses the points raised during the review process.

We look forward to receiving your revised manuscript.

Kind regards,

Nan Jiang

Academic Editor

PLOS ONE

2. Thank you for stating the following financial disclosure: [Rush University Medical Center]. At this time, please address the following queries: a) Please clarify the sources of funding (financial or material support) for your study. List the grants or organizations that supported your study, including funding received from your institution. b) State what role the funders took in the study. If the funders had no role in your study, please state: “The funders had no role in study design, data collection and analysis, decision to publish, or preparation of the manuscript.”c) If any authors received a salary from any of your funders, please state which authors and which funders.d) If you did not receive any funding for this study, please state: “The authors received no specific funding for this work.”Please include your amended statements within your cover letter; we will change the online submission form on your behalf.

Reviewers' comments:

Reviewer's Responses to Questions

**Comments to the Author**

1. Is the manuscript technically sound, and do the data support the conclusions?

Reviewer #1: Partly

Reviewer #2: Yes

2. Has the statistical analysis been performed appropriately and rigorously? 

Reviewer #1: Yes

Reviewer #2: Yes

3. Have the authors made all data underlying the findings in their manuscript fully available?

Reviewer #1: No

Reviewer #2: No

4. Is the manuscript presented in an intelligible fashion and written in standard English?

Reviewer #1: No

Reviewer #2: Yes

5. Review Comments to the Author

Reviewer #1: The manuscript entitled “Polymorphisms in rs2069845 are associated with IL-6 and soluble IL-6 receptor levels during total joint replacement” is a well-written, interesting topic, some modifications are needed.

1- The language/grammar/syntax needs to be edited in several areas especially the introduction and some of the methods as it was hard to understand in some areas.

2- Please italicize the gene throughout the manuscript.

3- What was the biological rationale for the selection of these SNPs?

4- Please insert a paragraph of sample size in the methodology section, please include the study power and a short description of how it was calculated.

5- How much is the MAF? Please include it in the method section

6- The Hardy-Weinberg equilibrium should be performed considering the control and subjects population

7- In discussion, please provide focused description on the effects of the SNPs

8- Please, move the paragraph of ethical approval with the IRB number from the acknowledgment section to the methodology section.

Reviewer #2: Manuscript Number PONE-D-24-46321

Authors of the manuscript entitled “Polymorphisms in rs2069845 are associated with IL-6 and soluble IL-6 receptor levels during total joint replacement” have duly evaluated the three SNPs present in the promoter region of IL-6 gene and checked the dynamics with circulating IL-6 and sIL-6r level in pre- & post-operative conditions. Overall, the manuscript seems technically good. However, I have the following comments on the manuscript

Introduction:

1) Line no. 62 “several cytokines …… late-stage biomarkers” authors should mention the other cytokines.

2) why rs2069845 has been used by the author (reported for leprosy)?

Materials and method:

1) What sample size was predicted when the authors performed power analysis? Please mention the test details, sample size, and version of the Quanto power calculation tool.

2) Three SNPs are mentioned in the line no. 85 of the introduction, but it has been changed to four SNPs in line no. 103.

3) Why age and gender-matched control has not been taken.

4) Line no. 116 - initials of Lead investigators are not necessary.

5) Rewrite IL6-R (line no. 132) as IL-6r

6) The author should include either an abbreviation table in the beginning or expand the acronyms used in the manuscript wherever it is mentioned for the First time for example, TKR (Line no. 157); and THR (Line no. 157).

General comments:

1) Are there SNPs reported from other cytokine genes related to bone resorption after TJR?

2) What are the other biomarkers/proteins that influence IL6/IL6 receptor binding?

3) Besides implant material, what other possible factors that cause intrinsic sensitivity of patients also need to be mentioned?

4) Legends for figures 1 and 2 should have the p-value.

5) Provide footnote for tables 2,3, 4, and 5

6) Authors need to give details of statistical tests in Tables for clarity

7) Limitations of the study should be incorporated after the discussion.

8) The authors need to improve the resolution of all the figures

9) Discussion and conclusion require a systematic writing

6. PLOS authors have the option to publish the peer review history of their article (what does this mean? ). If published, this will include your full peer review and any attached files.

**Do you want your identity to be public for this peer review?** For information about this choice, including consent withdrawal, please see our Privacy Policy .

Reviewer #1: No

Reviewer #2: No

---

## [Author Response · Author response to Decision Letter 1]

22 Jan 2025

We thank the reviewers for their helpful feedback. We have attempted to address each of their comments as best as possible and include a response to each, as well as any changes made in the manuscript in response.

---

## [Decision Letter · Decision Letter 1]

27 Feb 2025

PONE-D-24-46321R1Polymorphisms in rs2069845 are associated with IL-6 and soluble IL-6 receptor levels during total joint replacementPLOS ONE

Dear Dr. Ross,

Thank you for submitting your manuscript to PLOS ONE. After careful consideration, we feel that it has merit but does not fully meet PLOS ONE’s publication criteria as it currently stands. Therefore, we invite you to submit a revised version of the manuscript that addresses the points raised during the review process.

We look forward to receiving your revised manuscript.

Kind regards,

Nan Jiang

Academic Editor

PLOS ONE

Reviewers' comments:

Reviewer's Responses to Questions

**Comments to the Author**

1. If the authors have adequately addressed your comments raised in a previous round of review and you feel that this manuscript is now acceptable for publication, you may indicate that here to bypass the “Comments to the Author” section, enter your conflict of interest statement in the “Confidential to Editor” section, and submit your "Accept" recommendation.

Reviewer #1: All comments have been addressed

Reviewer #2: All comments have been addressed

2. Is the manuscript technically sound, and do the data support the conclusions?

Reviewer #1: Yes

Reviewer #2: Partly

3. Has the statistical analysis been performed appropriately and rigorously? 

Reviewer #1: Yes

Reviewer #2: Yes

4. Have the authors made all data underlying the findings in their manuscript fully available?

Reviewer #1: Yes

Reviewer #2: Yes

5. Is the manuscript presented in an intelligible fashion and written in standard English?

Reviewer #1: Yes

Reviewer #2: Yes

6. Review Comments to the Author

**Reviewer #1: ** Thanks alot for your efforts, in doing all required modifications

All comments have been addressed, and the manuscript is valid or publication in the current form

**Reviewer #2:**  All necessary corrections have been addressed by the authors. Nonetheless, I have a few additional suggestions that will further improve the manuscript.

1. In line 153, there is a typo that needs to be corrected. Please change "bdSNP" to "dbSNP.".

2. The minor allele frequency in line 154 should be rechecked using the 1000 Genomes database or dbSNP for the European population rather than the global population, followed by SNP analysis. For instance, the ref. allele “C” of rs4537545 has a frequency of 0.592611 in the European population (rs4537545 RefSNP Report - dbSNP - NCBI), whereas in figure-3, very few individuals are homozygous CC, with low mean pre-operative IL-6 level. Therefore, normally, SNP-based studies or analyses should not be done with a low sample size.

3. What tests are used for Hardy-Weinberg equilibrium analysis of SNPs in R-Studio? Please mention the test name in the methodology section.

4. The SNPs discussed in the article are located within introns (rs4537545 RefSNP Report - dbSNP - NCBI ; rs2069845 RefSNP Report - dbSNP - NCBI) and missense variant (rs2228145 RefSNP Report - dbSNP - NCBI), therefore, to strengthen these findings, the authors should conduct in-silico analyses to demonstrate how these SNPs regulate IL6 levels.

5. Authors can utilize various online tools available on the UCSC browser for this purpose. This approach would also differentiate their manuscript from previous studies focusing on these variants (e.g., reference 25).

7. PLOS authors have the option to publish the peer review history of their article (what does this mean? ). If published, this will include your full peer review and any attached files.

**Do you want your identity to be public for this peer review?** For information about this choice, including consent withdrawal, please see our Privacy Policy .

Reviewer #1: **Yes: ** Sally Mohammed El-Hefnawy

Reviewer #2: No

---

## [Author Response · Author response to Decision Letter 2]

24 Mar 2025

We thank the reviewers for their continued support. We have updated the manuscript per reviewer 2's recommendations and believe we have addressed all their concerns.

---

## [Decision Letter · Decision Letter 2]

3 Apr 2025

Polymorphisms in rs2069845 are associated with IL-6 and soluble IL-6 receptor levels during total joint replacement

PONE-D-24-46321R2

Dear Dr. Ross,

We’re pleased to inform you that your manuscript has been judged scientifically suitable for publication and will be formally accepted for publication once it meets all outstanding technical requirements.

Kind regards,

Nan Jiang

Academic Editor

PLOS ONE

Reviewers' comments:

Reviewer's Responses to Questions

**Comments to the Author**

1. If the authors have adequately addressed your comments raised in a previous round of review and you feel that this manuscript is now acceptable for publication, you may indicate that here to bypass the “Comments to the Author” section, enter your conflict of interest statement in the “Confidential to Editor” section, and submit your "Accept" recommendation.

Reviewer #2: All comments have been addressed

2. Is the manuscript technically sound, and do the data support the conclusions?

Reviewer #2: Yes

3. Has the statistical analysis been performed appropriately and rigorously? 

Reviewer #2: Yes

4. Have the authors made all data underlying the findings in their manuscript fully available?

Reviewer #2: Yes

5. Is the manuscript presented in an intelligible fashion and written in standard English?

Reviewer #2: Yes

6. Review Comments to the Author

Reviewer #2: Authors have made an effort to include all the suggested comments. The Manucsript is now technically sound and meets the standards of Plos One journal, so it can be accepted for publication.

7. PLOS authors have the option to publish the peer review history of their article (what does this mean? ). If published, this will include your full peer review and any attached files.

**Do you want your identity to be public for this peer review?** For information about this choice, including consent withdrawal, please see our Privacy Policy .

Reviewer #2: No

---

## [Editor Report · Acceptance letter]

PONE-D-24-46321R2

PLOS ONE

Dear Dr. Ross,

I'm pleased to inform you that your manuscript has been deemed suitable for publication in PLOS ONE. Congratulations! Your manuscript is now being handed over to our production team.

Kind regards,

on behalf of

Dr. Nan Jiang

Academic Editor

PLOS ONE